# Clinical and Imaging Features of COVID-19-Associated Pulmonary Aspergillosis

**DOI:** 10.3390/diagnostics12051201

**Published:** 2022-05-11

**Authors:** Tim Fischer, Yassir El Baz, Nicole Graf, Simon Wildermuth, Sebastian Leschka, Gian-Reto Kleger, Urs Pietsch, Manuel Frischknecht, Giulia Scanferla, Carol Strahm, Stephan Wälti, Tobias Johannes Dietrich, Werner C. Albrich

**Affiliations:** 1Division of Radiology and Nuclear Medicine, St. Gallen Cantonal Hospital, 9007 St. Gallen, Switzerland; yassir.elbaz@kssg.ch (Y.E.B.); simon.wildermuth@kssg.ch (S.W.); sebastian.leschka@kssg.ch (S.L.); stephan.waelti@kssg.ch (S.W.); tobias.dietrich@kssg.ch (T.J.D.); 2Clinical Trials Unit, St. Gallen Cantonal Hospital, 9007 St. Gallen, Switzerland; nicole.graf@kssg.ch; 3Division of Intensive Care, St. Gallen Cantonal Hospital, 9007 St. Gallen, Switzerland; gian-reto.kleger@kssg.ch; 4Department of Anesthesia, Intensive Care, Emergency and Pain Medicine, St. Gallen Cantonal Hospital, 9007 St. Gallen, Switzerland; urs.pietsch@kssg.ch; 5Division of Infectious Diseases and Hospital Epidemiology, St. Gallen Cantonal Hospital, 9007 St. Gallen, Switzerland; manuel.frischknecht@kssg.ch (M.F.); giulia.scanferla@kssg.ch (G.S.); carol.strahm@kssg.ch (C.S.); werner.albrich@kssg.ch (W.C.A.)

**Keywords:** COVID-19, COVID-19-associated pulmonary aspergillosis, chest CT

## Abstract

Background: COVID-19 superinfection by Aspergillus (COVID-19-associated aspergillosis, CAPA) is increasingly observed due to increased awareness and use of corticosteroids. The aim of this study is to compare clinical and imaging features between COVID-19 patients with and without associated pulmonary aspergillosis. Material and Methods: In this case–control study, hospitalized patients between March 2020 and March 2021 were evaluated. Two observers independently compared 105 chest CTs of 52 COVID-19 patients without pulmonary aspergillosis to 40 chest CTs of 13 CAPA patients. The following features were evaluated: lung involvement, predominant main pattern (ground glass opacity, crazy paving, consolidation) and additional lung and chest findings. Chronological changes in the abnormal extent upon CT and chronological changes in the main patterns were compared with mixed models. Patient-wise comparisons of additional features and demographic and clinical data were performed using Student’s t-test, Chi-squared test, Fisher’s exact tests and Wilcoxon rank-sum tests. Results: Compared to COVID-19 patients without pulmonary aspergillosis, CAPA patients were older (mean age (±SD): 70.3 (±7.8) versus 63.5 (±9.5) years (*p* = 0.01). The time-dependent evolution rates for consolidation (*p* = 0.02) and ground glass (*p* = 0.006) differed. In early COVID-19 disease, consolidation was associated with CAPA, whereas ground glass was less common. Chronological changes in the abnormal extent upon CT did not differ (*p* = 0.29). Regardless of the time point, bronchial wall thickening was observed more frequently in CAPA patients (*p* = 0.03). Conclusions: CAPA patients showed a tendency for consolidation in early COVID-19 disease. Bronchial wall thickening and higher patient age were associated with CAPA. The overall lung involvement was similar between both groups.

## 1. Introduction

The COVID-19 pandemic caused by SARS-CoV-2 has challenged health-care systems due to the introduction of a new virus without population immunity and subsequently new viral mutations with increased transmissibility or immune escape [1]. Already during the first wave, a large proportion of hospitalized patients required prolonged hospitalization and intensive care treatment. COVID-19 is a multisystem disorder with a complex pathophysiology [2]. In the lungs, SARS-CoV-2 causes diffuse alveolar damage. Particularly beyond the early viral replication phase, which is characterized by a viral syndrome, lung damage in severe COVID-19 is caused by immunothrombosis [2]. In acute respiratory failure, non-invasive support with close monitoring is now widely accepted as opposed to early intubation at the pandemic onset [3]. Increased risk for a fatal outcome includes older age, male gender, immunosuppression and cardiovascular or metabolic diseases [4]. Chest-computed tomography has quickly become a standard complementary technique alongside rt-PCR for establishing a diagnosis and for risk assessment [5].

On imaging, lung abnormalities tend to start with ground glass opacity (GGO), followed by a crazy paving pattern and consolidation [6,7,8], typically peaking at days 9 to 13 after symptom onset [6,8].

The use of high-doses corticosteroids in patients hospitalized with COVID-19 has been shown to have a positive effect on patient survival, although the immunosuppressant effect leads to an increasing risk of superinfection by Aspergillus spp. (COVID-19-associated aspergillosis, CAPA) [9,10], which is reported with an overall incidence among COVID-19 patients of 13.5% (range 2.5 to 35.0%) [11]. Consensus criteria for case definitions of CAPA have been proposed by the European Confederation for Medical Mycology (ECMM) and the International Society for Human and Animal Mycology (ISHAM). Since there is no established consensus for prophylaxis, early recognition of CAPA is essential for patient management, meaning better radiographic diagnosis is desirable.

Three different grades were proposed: possible, probable and proven CAPA [10]. CAPA is reported in 1% to 28% of intensive care unit (ICU) patients [12,13,14,15], with mortality rates ranging between 44 and 71% [15,16,17].

Imaging features have mostly been reported as case-based without control groups. Consolidations and nodules were described as predominant CT findings, whereas cavities and halo signs are rare in contrast with patients with neutropenia-associated invasive pulmonary aspergillosis [10,18,19,20,21,22,23]. A recent study found a potential association between CAPA and bronchiectasis [24].

This is the first study aiming to establish typical CAPA imaging features by comparing CT scans of CAPA patients to scans of COVID-19 patients without evidence of pulmonary aspergillosis.

## 2. Materials and Methods

This was a case–control study nested in a prospective cohort study of critically ill patients with COVID-19. IRB approval by the local ethics committee and informed consent were obtained from all evaluated patients. Evaluated patients were hospitalized between March 2020 and March 2021. CAPA patients were identified from the infectious disease consultation database and were hospitalized with a diagnosis of CAPA at the St. Gallen Cantonal Hospital, with the requirement for high-level care (ICU or a specialized pulmonary unit). Only patients with SARS-CoV-2 confirmed by rt-PCR or antigen assay were included. Pulmonary aspergillosis was diagnosed by fungal cultures performed on bronchoalveolar lavage (BAL); tracheal secretions or sputum and galactomannan (GM) testing (Enzyme immune assay, Biorad, Hercules, CA, USA) performed on BAL; or tracheal secretions, sputum or serum. CAPA was categorized as possible, probable or proven, based on recent consensus guidelines (ECMM/ISHAM), including recommended cut-off values for GM [10]. Patients were also evaluated for tracheal aspergillosis; in no case were criteria for tracheal aspergillosis fulfilled.

As controls, we used patients with rt-PCR or antigen-confirmed COVID-19 without a CAPA diagnosis (subsequently referred to as COVID-19 patients) who were admitted to the ICU and were enrolled at the St. Gallen Cantonal Hospital in the prospective multicenter study (CRiPSI: COVID-19 Risk Prediction in Swiss ICUs-Trial). These patients underwent at least weekly screening for pulmonary aspergillosis while intubated with tracheal secretions or non-bronchoscopy lavage using fungal cultures and GM testing. In addition, fungal cultures and GM testing were also performed on samples of clinically indicated BAL. Serum GM was not routinely assessed and was mainly measured in patients with suspected CAPA.

For both groups, the following inclusion criteria applied: signed informed consent and age ≥18 years. The following exclusion criteria were applied: documented refusal to participate in the study or COVID-19 diagnosis after ICU discharge.

### 2.1. Imaging and Image Evaluation

All imaging was performed in our institution’s Siemens Somatom Force or Somatom Flash device (Siemens Healthineers, Erlangen, Germany). Patients were scanned in prone position. Somatom Force settings: 100 kV and 100 mAs with tin filter; Somatom Flash settings: 100 kV and 50 mAs. If CT angiography was required to check for pulmonary embolism, patients were scanned in supine position with an injection of 40 mL iodine-based contrast agent (Xenetix 350, Guerbet AG, Villepinte, France, flow 4 mL/s), following a test injection with 15 mL contrast agent with the following settings: Somatom Force: 100 kV and 105 mAs automated dose modulation (care kV and Care Dose 4D) enabled; Somatom Flash: 100 kV and mAs 110 automated dose modulation (care kV and Care Dose 4D) enabled. Reconstructions were performed using an iterative algorithm: axial 5 mm and 2 mm slices (standard kernel and bone kernel) and sagittal 5 mm slices of the spine (bone kernel). Imaging was performed as clinically indicated according to the treating physicians.

Two radiologists (TF and YEB) with seven and eight years of experience in radiology individually evaluated all CT examinations from all included patients, blinded from all clinical and microbiological information. Estimations of lung involvement have been performed in previous studies [8,25]. In this study, lobe-wise estimations (right upper lobe, middle lobe, right lower lobe, left upper lobe and left lower lobe) of affected lung tissue were conducted as percentages (%). For each CT examination, a subsequent weighted mean for the total lung involvement was calculated through the use of previously described proportions [26]. Furthermore, involvement of the upper lobe and lower lobe was calculated likewise (middle lobe involvement was attributed to the upper lobe).

The main pattern was categorized as ground glass opacity (GGO) (region of increased lung attenuation in which vessels remain visible), crazy paving (appearance of ground-glass opacities with superimposed interlobular and intralobular septal thickening) or consolidation (homogeneous increased lung opacity with obscuration of vessels).

Additional findings of COVID-19-related imaging features were based on a meta-analysis of 19 retrospective studies [27] and evaluated as present or absent: subpleural linear opacity, septal thickening, subpleural reticulation, air bronchogram, pleural thickening, bronchiectasis, bronchial wall thickening, tree-in-bud and vascular enlargement. Intrathoracic lymph node enlargement, pleural effusion (if bilateral, both sides were summed up) and pericardial effusion were measured in millimeters. For the evaluation of bronchial wall thickness, the T/D ratio (wall thickness (T) divided by total diameter of bronchus (D)) was estimated. Values between 0.1 to 0.2 have been described as normal [28,29]. In this study, a value above 0.2 was considered to represent bronchial wall thickening. As previously described, case-based imaging features of CAPA were evaluated in all patients as present or absent based on the literature: pulmonary nodules [18,19,20,23], cavitation [19,23], halo sign [21] and reverse halo sign [22]. Tracheal thickening or calcifications have been reported as imaging findings in invasive tracheal aspergillosis. In the context of influenza (influenza associated aspergillosis, IAPA), manifestations include airway plaques, pseudomembranes or ulcerations [10,30,31,32]. The trachea was evaluated for thickening, calcification and wall irregularities, which could represent some of the above-mentioned pathologies. The findings were summarized as tracheal abnormalities and as potential direct signs of the tracheobronchial manifestation of CAPA [10] and evaluated as present or absent. The abnormal content of the lumen, e.g., mucus, was not considered a tracheal abnormality. Previously published imaging examples served as a reference for evaluation of the main pattern and the additional chest findings [33]. The presence of emphysema was evaluated as a potential pre-existing condition.

### 2.2. Evaluation of Clinical Features

The following pre-existing clinical conditions of included patients were reviewed in the medical database. Established diagnoses were based on current guidelines: chronic obstructive pulmonary disease (COPD), asthma, neoplasm, hematologic disease, diabetes, hypertension, cardiovascular disease, cerebrovascular disease or autoimmune disease. The following medications prior to the COVID-19 disease were reviewed: prior use of steroids and prior use of immunosuppressive drugs. The following treatments during COVID-19 disease were reviewed: use of steroids, use of immunomodulating drugs and antiviral treatment. The following complicating factors during COVID-19 disease were reviewed: bacterial superinfection and renal failure. Bacterial superinfections were either proven or suspected by infectious disease experts. Cause of death was reviewed and categorized as death from acute respiratory distress syndrome (ARDS), COVID-19 pneumonia or multi organ failure.

### 2.3. Statistics

Statistical analysis was performed on SPSS version 25 (IBM Corp, Armonk, NY, USA) or in the R programming language (version 4.0.2) [34] by NG. The package “nlme” [35] was used to compute the linear mixed-effects model. The package “lme4” [36] was used to compute the logistic mixed-effects model. The package “tableone” [37] was used for descriptive statistics and to calculate Fisher’s exact test and the Wilcoxon rank sum test. The package “ggeffects” [38] was used to compute the predicted probabilities. The package “ggplot2” [39] was used to plot the figures.

Baseline characteristics were reported as means and standard deviations (SD) or median and interquartile ranges (IQRs) for continuous data or as absolute and relative frequencies for categorical variables, and were compared with Student’s t-test for independent samples and the Chi-squared test.

The inter-rater reliability of chest CT double readouts was tested with the intraclass correlation coefficient (ICC) using single measures of the mixed-effects model with absolute agreement (continuous variables: lobe involvement, pleural effusion, lymphadenopathy and pericardial effusion). Values >0.5 were considered moderate, >0.75 good and >0.9 very good agreement [40] or Cohen’s Kappa (categorical variables: all other variables). Values >0.4 were considered moderate, >0.6 substantial and >0.8 almost perfect agreement [41].

## 3. Results

In total, 54 COVID-19 patients and 15 COVID-19-associated pulmonary aspergillosis patients were potentially eligible. Three patients had no chest CT and one patient refused informed consent, while a total of 52 COVID-19 patients and 13 CAPA patients with 145 CT examinations (105 COVID-19, 40 CAPA) were evaluated. Pulmonary aspergillosis was diagnosed via fungal cultures performed on pleural biopsy (*n* = 1), BAL (*n* = 4), tracheal secretions (*n* = 1) and GM testing performed on BAL (*n* = 6). In addition, a good-quality sputum with a galactomannan index ≥4.5 was allowed in the absence of a bronchoscopy as a mycological criterion for possible CAPA (*n* = 1). CAPA diagnosis was proven (*n* = 1), probable (*n* = 10) or possible (*n* = 2).

### 3.1. Baseline Characteristics

Key demographics are given in Table 1. In contrast to CAPA, there were 54% (7/13) males among patients with CAPA versus 79% (41/52) among patients with COVID-19 (*p* = 0.07). CAPA patients were older compared to COVID-19 patients (*p* = 0.01).

Overall, 43% (28/65) of all COVID-19 and CAPA patients did not survive, and mortality rates were higher in CAPA patients (62%, 8/13) compared to COVID-19 (39%, 20/52), but without statistical significance (*p* = 0.13). In the 8 deceased CAPA patients, the leading cause of death was ARDS (46%, 6/13), followed by death due to COVID-19 pneumonia (15%, 2/13). In the 20 deceased COVID-19 patients, the leading cause of death was ARDS (23%, 12/52), followed by death due to COVID-19 pneumonia (14%, 7/52), while one patient died from multi organ failure (2%, 1/52, *p* = 0.34).

The median time from symptom onset until detection of SARS-CoV-2 was 3 days (IQR 6 days, range 3 days prior to symptom onset to 14 days after symptom onset). The median time from symptom onset until CAPA diagnosis was 16 days (IQR 16 days, range 0 to 42 days). The median time from detection of SARS-CoV-2 until the diagnosis of CAPA was 10 days (IQR 12 days, range 0 to 35 days). Overall, the median time from symptom onset to ICU admission was 9 days (IQR 5.75 days, range 2 to 42 days).

### 3.2. Comparison of Lung Involvement

The lobe-wise inter-rater agreement (ICC) was very good (range: 0.87 to 0.98). Complete involvement and lobe-wise involvement (upper lobe and lower lobe) by days after symptom onset for COVID-19 patients and CAPA patients are given in Figure 1. Median times to detection for SARS-CoV-2, ICU admission and CAPA diagnosis are overlaid.

The results of the random intercept and random slope models confirmed that lung involvement is overall time-dependent, but chronological changes in the abnormal extent of CT did not differ between COVID-19 and CAPA patients (*p* = 0.29 for interaction) (Appendix A).

### 3.3. Comparison of Main Pattern

The inter-rater agreement (Cohen’s kappa) was substantial (0.78).

#### 3.3.1. Consolidation

Consolidation as a main pattern over time differed between COVID-19 patients and CAPA patients. Figure 2 shows predicted probabilities by days after onset of symptoms. Predicted probabilities of consolidation were higher for CAPA patients, especially in early (up to 5 days after symptom onset) disease, and decreased over time, while the probability of consolidation increased over time for COVID-19 patients. Thus, the chronological changes in consolidation were different between CAPA and COVID-19 patients (*p* = 0.02 for interaction) (Appendix A).

#### 3.3.2. Crazy Paving

Predicted probabilities of crazy paving as a main pattern increased over time for both types of patients. Figure 2 shows predicted probabilities by days after onset of symptoms (*p* = 0.18 for interaction) (Appendix A).

#### 3.3.3. Ground Glass Opacity

GGO as a main pattern over time differed between COVID-19 and CAPA patients. Figure 2 shows predicted probabilities by days after onset of symptoms. GGO was a very typical pattern for COVID-19 patients in early (up to 5 days after symptom onset) disease. For CAPA patients, the predicted probabilities of GGO as a predominant pattern were much lower early in the COVID-19 disease. The decrease in predicted probabilities over time was higher for patients without than with pulmonary aspergillosis (*p* = 0.006 for interaction) (Appendix A).

### 3.4. Comparison of Additional Findings

A comparison of additional findings between patients with COVID-19 and with CAPA (Table 2) indicated that bronchial wall thickening (*p* = 0.03) was observed more often in the CAPA group. Representative images of chest CTs of a patient with CAPA are shown in Figure 3.

### 3.5. Comparison Clinical Features

The results obtained for the reviewed clinical features are given in Table 3. Except for therapy with immunomodulating drugs during COVID-19 disease (*p* = 0.04), no differences were observed between COVID-19 and CAPA patients. The immunomodulating drugs were chloroquine and tocilizumab. The antiviral therapy was remdesivir, while the steroid therapy was dexamethasone.

### 3.6. Bacterial Superinfection as a Potential Confounder in CAPA Imaging Features

In approximately 50% of COIVD-19 patients, a bacterial superinfection was proven or suspected. Bacterial superinfection may be a confounding variable when imaging findings in COVID-19 and CAPA are compared; thus, a subgroup analysis for the key findings was conducted, taking this consideration into account. Bronchial wall thickening was observed in 61.5% (8/13) of CAPA patients, in 33.3% (9/27) of COIVD-19 patients with bacterial superinfection and in 20.0% (5/25) of COVID-19 patients without bacterial superinfection, and the differences were significant (*p* = 0.04).

The same subgroup analysis was conducted for the time-dependent probability of consolidation between the three groups (Appendix A, Figure 4). In early disease, compared to COVID-19 patients without bacterial superinfection, consolidation is slightly more common in COVID-19 patients with bacterial superinfection, and most common in CAPA patients.

## 4. Discussion

This case–control study compared clinical and imaging features between COVID-19 patients and patients with COVID-19-associated pulmonary aspergillosis (CAPA). CAPA patients were older compared to COVID-19 patients (*p* = 0.01). Consolidation in early COVID-19 disease was associated with CAPA (*p* = 0.02) but overall lung involvement over times did not differ between both groups (*p* = 0.29). Regardless of the time point, bronchial wall thickening was associated with CAPA (*p* = 0.03). The use of immunomodulating drugs during COVID-19 disease may be a risk factor for developing CAPA (*p* = 0.04), although due to the low numbers a general conclusion has to be drawn with care.

The imaging features described in this study support physicians in diagnosing CAPA and may lead to improved patient management.

The age distribution in this study reflects the findings from previous studies. In a meta-analysis including 186 CAPA patients (97.8% hospitalized on an ICU), the median age was reported as 68 years (70 years in this study) [15].

Unlike the previously reported male predominance of 73% in the same meta-analysis [15], in our data differences in gender were not observed for patients with CAPA. Due to the small sample size with subsequent large error margins, conclusions about epidemiologic characteristics of our patient population should be made only reluctantly.

The reported CAPA mortality in our study (62%) was similar to that reported in the literature: 50–60% [15], 60–70% [12] or 44–71% [17] and higher compared to our COVID-19 patients without CAPA. The lack of a statistical difference in our study indicates the critical ill patient status of our control group.

Lung involvement was reported to peak at around 9 to 13 days after symptom onset in patients with COVID-19 without pulmonary aspergillosis [6]. In our study, involvement tended to peak later after symptom onset (approximately day 20 to 25). To the best of our knowledge, there are no available studies describing the chronological changes in the abnormal extent of CT in critically ill patients. This indeed makes our study unique and important. In a large cohort study of critically ill patients, which did not report radiologic data [42], ICU admission occurred 8 days after symptom onset, with a median ICU stay of 12 days, which may indicate a prolonged disease compared to patients without ICU requirement. Since only critically ill patients were evaluated in our study, this may explain the later peak and prolonged lung involvement compared to established COVID-19 stages that encompass a more general population. Moreover, lung involvement has been shown to remain high until at least day 24 after the peak stage has been reached between days 6 and 11 [7].

Chest CT findings in COVID-19 usually evolve from GGO to crazy paving and consolidation [6,7,8,43], which is supported by this study. Unlike in COVID-19 without aspergillosis, consolidation was more prevalent in early disease in CAPA patients, where it represents alveolar infiltration of inflammatory cells, which is a later finding in COVID-19 (without aspergillosis). Early COVID-19 is rather characterized by predominantly ground lass opacities typical of viral infections. It is, therefore, essential to consider the timing of this finding. More consolidation than expected in the distinct early stage of the disease may be indicative of aspergillus superinfection.

Neither the previously described case-based CAPA features, such as nodularity [18,20,21,22,23], nor features typical of neutropenia-associated invasive pulmonary aspergillosis, such as halo sign [44], reverse halo sign [22] and cavitation [18,23], were observed more frequently compared to in COVID-19 patients. Bacterial superinfection in COVID-19 patients was addressed as a potential confounding variable, with the results suggesting that bronchial wall thickening and early consolidation are more commonly observed in CAPA patients compared to COVID-19 patients without bacterial superinfection, and also to a lesser extent compared to patients with bacterial superinfection. It may be concluded that both features are suggestive of CAPA. We also performed a subgroup analysis to compare CT examinations after an established CAPA diagnosis with CT examinations of COVID-19 patients without pulmonary aspergillosis that were performed in a corresponding time frame (median time from COVID-19 to CAPA diagnosis was 10 days, compared CTs were performed ≥10 days after COVID-19 diagnosis). This subgroup analysis also did not reveal the previously described typical imaging features (data not shown). One explanation may be that this study only included critically ill patients with extensive pulmonary disease, where findings such as halo or reverse halo signs may have been masked. Alternatively, COVID-19 as a predisposing condition leads to less epithelial damage than influenza or neutropenia, which is confirmed by the lower prevalence of positive serum galactomannan in CAPA as a surrogate for more mucosal than angioinvasive disease [16]. Bronchial wall thickening as a marker of airway infection has been described as a feature of aspergillosis [30,32]. Due to the usually extensive COVID-19-associated lung alterations, bronchial wall thickening may be the only evidence of aspergillus superinfection, particularly since we did not observe any signs of tracheal involvement. Regarding clinical data, except for treatment with immunomodulating drugs, COVID-19 and CAPA patients did not differ. A higher incidence of CAPA in patients under immunomodulating therapy is plausible, but the observed numbers are low and a general conclusion has to be drawn with caution. The lack of a statistical difference with regard to the use of steroids likely relates to both the low number of CAPA patients receiving steroids prior to admission, while the application of steroids during COVID-19 disease likely was a more important risk factor for CAPA, but this was almost universal and did not differ between patients with CAPA and without CAPA.

This study had the following strengths. First, groups of patients were well-characterized with systematic screening for CAPA. Second, the image evaluation was performed systematically by two experienced radiologists with very good inter-rater reliability. Third, imaging features were not assigned to different disease stages based on arbitrary time windows. Instead, imaging features were analyzed with mixed models that considered the correlated nature of the data, with many patients having more than one CT.

This study had the following limitations. First, some patients may have died during early COVID-19 disease before CAPA had the opportunity to manifest (competing risks). In this study population, only one patient died early (<7 days) after ICU admission, meaning the statistical effect may be negligible. Second, the number of CAPA patients was lower compared to COVID-19 patients. Third, in view of the multiplicity of tests, the results from additional features have to be interpreted with caution.

## 5. Conclusions

In conclusion, patients with COVID-19-associated pulmonary aspergillosis showed a tendency for consolidation in early COVID-19 disease. Bronchial wall thickening and higher patient age were associated with CAPA. Prior use of immunosuppressive drugs may be a risk factor. Previously reported findings for COVID-19-associated aspergillosis or invasive pulmonary aspergillosis (nodularity, halo sign, reverse halo sign or cavitating disease) were not observed. This is the first study of its kind and the findings may assist physicians in raising suspicion of the disease.

## Figures and Tables

**Figure 1 diagnostics-12-01201-f001:**
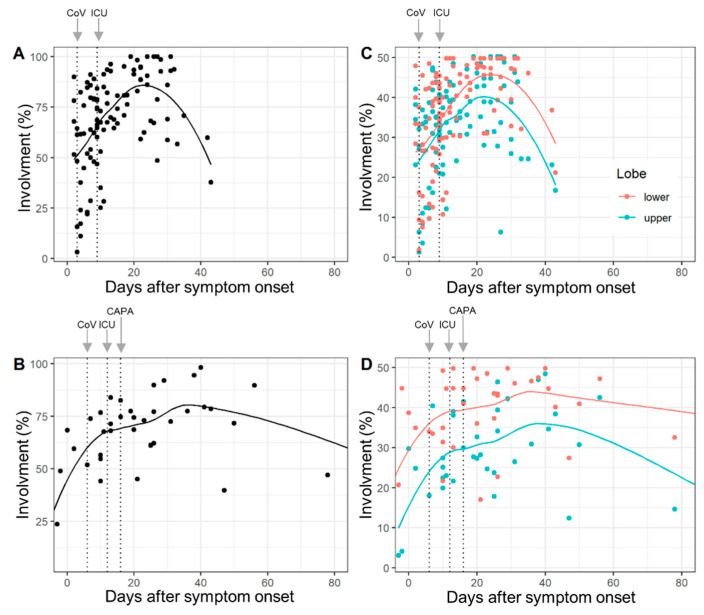
Total lung involvement (**A**,**B**) and lobe-wise involvement (**C**,**D**) (upper lobe: green line, lower lobe: red line) by days after COVID-19 symptom onset in COVID-19 patients (**A**,**C**) and COVID-19-associated pulmonary aspergillosis (CAPA) patients (**B**,**D**). Dots show % of affected tissue of individual CTs and the curved lines represent the locally estimated scatterplot smoothing (loess) regression line. The dotted vertical lines (left to right) illustrate median days until detection of SARS-CoV-2 (CoV), ICU admission (ICU) and CAPA diagnosis (CAPA). Chronological changes in the abnormal extent of CT did not differ between COVID-19 and CAPA patients (*p* = 0.29) for interactions.

**Figure 2 diagnostics-12-01201-f002:**
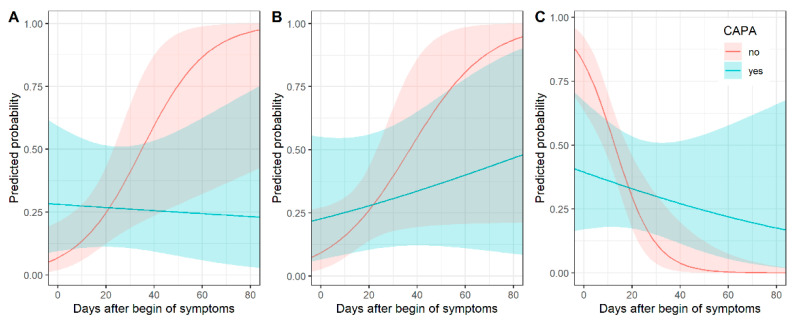
Chronological changes in CT (predicted probabilities) of consolidation (**A**), crazy paving (**B**) and ground glass opacity (**C**) for patients with and without COVID-19-associated pulmonary aspergillosis (CAPA) with corresponding 95% confidence intervals. In early COVID-19 disease, consolidation was associated with CAPA, whereas ground glass opacity was less common.

**Figure 3 diagnostics-12-01201-f003:**
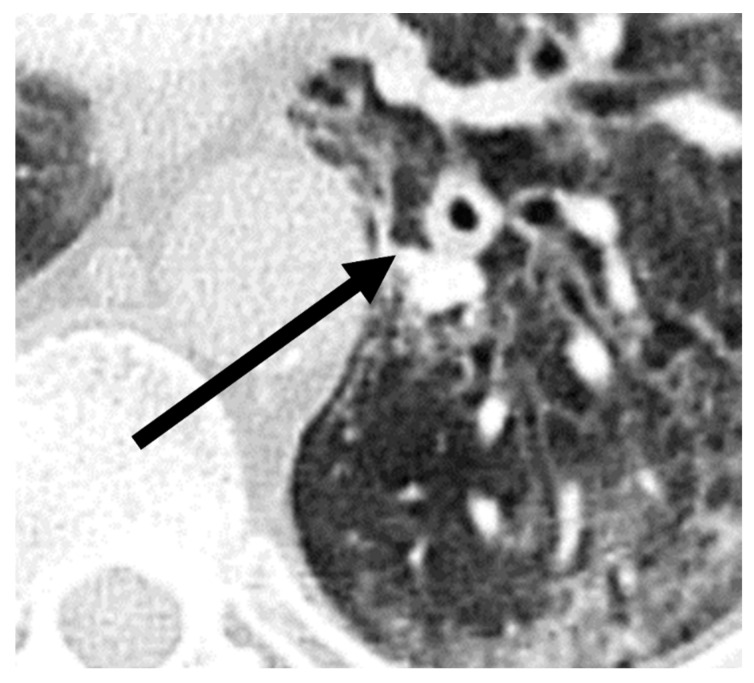
Imaging example of bronchial wall thickening (arrow) in a patient with probable COVID-19-associated pulmonary aspergillosis (CAPA). CT was performed 10 days after symptom onset on the day of the CAPA diagnosis. The T/D ratio (wall thickness (T) divided by the total diameter of bronchus (D)) was 0.32 in this case.

**Figure 4 diagnostics-12-01201-f004:**
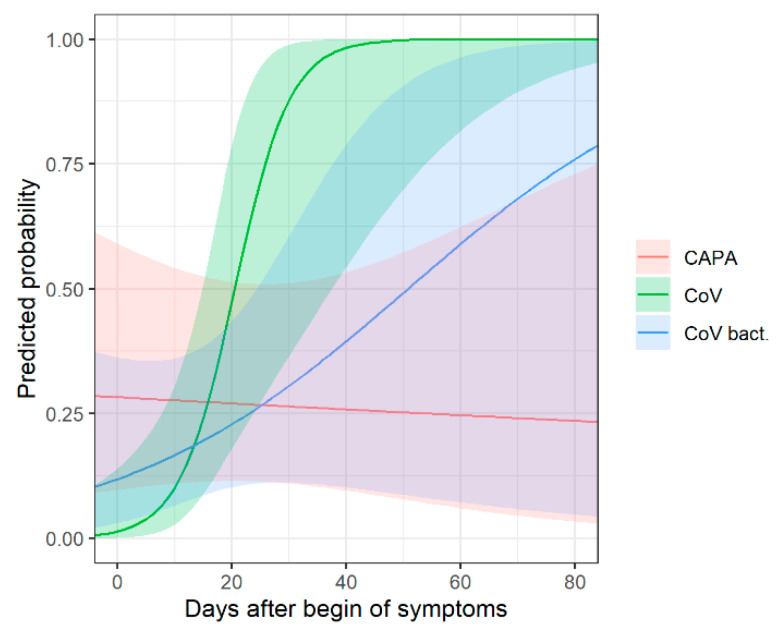
Chronological changes in CT (predicted probabilities of consolidation) with corresponding 95% confidence intervals for patients with COVID-19-associated pulmonary aspergillosis (CAPA), COVID-19 without bacterial superinfection (CoV) and COVID-19 with bacterial superinfection (CoV bact.). In early disease, consolidation is most common in CAPA patients and more common in COVID-19 patients with bacterial superinfection compared to patients without bacterial superinfection.

**Table 1 diagnostics-12-01201-t001:** Key demographics.

	Overall (*n* = 65)	COVID-19 (*n* = 52)	CAPA * (*n* = 13)	*p*-Value
Mean age in years (±SD)	64.8 (±9.5)	63.5 (±9.5)	70.3 (±7.8)	**0.01**
Gender				0.07
male	74% (48/65)	79% (41/52)	54% (7/13)	
female	26% (17/65)	21% (11/52)	46% (6/13)	

* CAPA–COVID-19-associated pulmonary aspergillosis.

**Table 2 diagnostics-12-01201-t002:** Comparison of additional findings between patients with COVID-19 and with CAPA.

Variable	COVID-19	CAPA *	*p*-Value	Cohen’s Kappa/ICC
Subpleural linear opacity	25.0% (13/52)	30.8% (4/13)	0.73	0.97
Septal thickening	65.4% (34/52)	61.5% (8/13)	1.00	0.90
Subpleural reticulation	17.3% (9/52)	23.1% (3/13)	0.69	0.77
Air bronchogram	63.5% (33/52)	69.2% (9/13)	0.76	0.83
Pleural thickening	9.6% (5/52)	15.4% (2/13)	0.62	1
Halo sign	13.5% (7/52)	0% (0/13)	0.33	0.83
Reverse halo sign	0% (0/52)	0% (0/13)	NA	NA
Bronchiectasis	19.2% (10/52)	38.5% (5/13)	0.16	0.92
Bronchial wall thickening	26.9% (14/52)	61.5% (8/13)	**0.03**	0.89
Tree in bud	5.8% (3/52)	0% (0/13)	1.000	1
Cavitating lung lesions	11.5% (6/52)	23.1% (3/13)	0.37	0.96
Pulmonary nodules	3.8% (2/52)	7.7% (1/13)	0.49	0.56
Vascular enlargement	19.2% (10/52)	30.8% (4/13)	0.45	0.66
Trachea abnormal ^a^	7.7% (4/52)	7.7% 1/13	1.00	0.66
Emphysema	7.7% (4/52)	0% (0/13)	0.58	0.34
Pleural effusion (median [IQR])	0.0 [0.0, 8.1]	4.5 [0.0, 16.7]	0.40	0.98
Lymphadenopathy (median [IQR])	7.9 [7.0, 9.8]	7.8 [7.1, 10.7]	0.69	0.83
Pericardial effusion (median [IQR])	0.0 [0.0, 0.5]	0.2 [0.0, 0.5]	0.31	0.93

* CAPA–COVID-19-associated pulmonary aspergillosis. ^a^ The following tracheal abnormalities were observed: CAPA group: wall irregularity (*n* = 1); COVID-19 group: tracheal nodules (*n* = 2), wall irregularity (*n* = 1) and tracheal diverticulum (*n* = 1).

**Table 3 diagnostics-12-01201-t003:** Comparison of pre-existing conditions, treatment during COVID-19 disease, prior medication and complication factors during COVID-19 disease.

Pre-Existing Condition	COVID-19	CAPA *	*p*-Value
COPD	5.8% (3/52)	15.4% (2/13)	0.26
Asthma	3.8% (2/52)	0% (0/13)	1.00
Other pulmonary disease ^a^	17.3% (9/52)	0% (0/13)	0.19
Neoplasm	5.8% (3/52)	7.7% (1/13	1.00
Hematologic disease	3.8% (2/52)	7.7% (1/13	0.49
Diabetes	38.5% (20/52)	30.8% (4/13)	0.75
Hypertension	42.3% (22/52)	61.5% (8/13)	0.23
Cardiovascular disease	26.9% (14/52)	46.2% (6/13)	0.20
Cerebrovascular disease	7.7% (4/52)	30.8% (4/13)	0.44
Autoimmune disease	0% (0/52)	0% (0/13)	-
**Prior medication**			
Prior use of steroids	7.7% (4/52)	6.2% (1/13)	1.0
Prior use of immunosuppressive drugs	3.8% (2/52)	7.7% (1/13)	0.49
**Treatment during COVID-19 disease**			
Use of steroids	98.1% (51/52)	100% (13/13)	1.00
Use of immunomodulating drugs	0% (0/52)	15.4% (2/13)	**0.04**
Antiviral treatment	13.5% (7/52)	7.7% (1/13)	1.00
**Complicating factors during COVID-19 disease**			
Bacterial superinfection (proven or suspected) ^b^	51.9% (27/52)	76.9% (10/13)	0.13
Renal failure	11.5% (6/52)	0% (0/13)	0.34

* CAPA–COVID-19-associated pulmonary aspergillosis. ^a^ Other pulmonary diseases include the following: obstructive sleep apnoea syndrome (*n* = 6), severe emphysema (*n* = 1), obesity hypoventilation syndrome (*n* = 1), atypical mycobacteriosis (*n* = 1). ^b^ Numbers of proven or suspected bacterial superinfections: COVID-19: 26 proven, 1 suspected, CAPA: 7 proven, 3 suspected.

## Data Availability

The data presented in this study are available on request from the corresponding author.

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
