# Peer review of "Clinical and Imaging Features of COVID-19-Associated Pulmonary Aspergillosis"

_diagnostics, 2022, doi:10.3390/diagnostics12051201_

Round 1

Reviewer 1 Report

  1. Introduction L37-42. The COVID-19 pandemic caused by SARS-CoV-2 is still challenging health-care systems due to new virus mutations with increased transmissibility and/or immune escape [1]. A large proportion of hospitalized patients require prolonged hospitalization and intensive care treatment. The case fatality ratio (defined as the proportion of individuals diagnosed with disease who die from that disease) is reported as 1-3% with higher risk among older patients with cardiovascular or metabolic diseases and men in general [2].  Could you please improve this paragraph and add this references:
  • Pelosi, P., Tonelli, R., Torregiani, C., Baratella, E., Confalonieri, M., Battaglini, D., Marchioni, A., Confalonieri, P., Clini, E., Salton, F., Ruaro, B. (2022). Different Methods to Improve the Monitoring of Noninvasive Respiratory Support of Patients with Severe Pneumonia/ARDS Due to COVID-19: An Update. Journal of clinical medicine11(6), 1704. https://doi.org/10.3390/jcm11061704
  • Asadi, F., Shahnazari, R., Bhalla, N., & Payam, A. F. (2021). Clinical evaluation of SARS-CoV-2 lung HRCT and RT-PCR Techniques: Towards risk factor based diagnosis of infectious diseases. Computational and structural biotechnology journal19, 2699–2707. https://doi.org/10.1016/j.csbj.2021.04.058
  1. L48-49. Use of high-doses corticosteroids in patients hospitalized with COVID-19 has been shown to have a positive effect on patient survival but the immunosuppressant effect  leads to increasing superinfection by Aspergillus spp (COVID-19 associated aspergillosis, CAPA) [6,7].

Please improve this paragraph and add these references:

a-Chong WH, Neu KP. Incidence, diagnosis and outcomes of COVID-19-associated pulmonary aspergillosis (CAPA): a systematic review. J Hosp Infect. 2021 Jul;113:115-129. doi: 10.1016/j.jhin.2021.04.012. Epub 2021 Apr 21. PMID: 33891985; PMCID: PMC8057923.

b-Baratella E, Roman-Pognuz E, Zerbato V, Minelli P, Cavallaro MFM, Cova MA, Luzzati R, Lucangelo U, Sanson G, Friso F, Bussani R, Pinamonti M, Busetti M, Salton F, Confalonieri M, Ruaro B, Di Bella S. Potential links between COVID-19-associated pulmonary aspergillosis and bronchiectasis as detected by high resolution computed tomography. Front Biosci (Landmark Ed). 2021 Dec 30;26(12):1607-1612. doi: 10.52586/5053. PMID: 34994174.

  1. L 59-61. Therefore, the aim of this study was to compare clinical and imaging features between CAPA patients and a well characterized group of critically ill COVID-19 patients without evidence of pulmonary aspergillosis. Please improve the description of study aim and underline the novelty of the study.
  2. 2.1. Imaging. Please summarise all the information regarding the imaging in the same paragraph.
  3. Table 1. Key demographic and clinical characteristics. Please add the most important clinical findings of patients.
  4. Figure 3. L259-261. Imaging example of bronchial wall thickening in a patient with probable COVID-19 associated pulmonary aspergillosis (CAPA). CT was performed 10 days after symptom onset at the day of the CAPA diagnosis. T/D ratio (wall thickness (T) divided by the total diameter of bronchus (D) was 0.32 in this case. Please improve the quality of figure 3.
  5. Discussion. L 297- 302. This case-control study compared clinical and imaging features between COVID-19 patients and patients with COVID-19 associated pulmonary aspergillosis (CAPA). CAPA  patients were older compared to COVID-19 patients (p=0.01). Consolidation in early  COVID-19 disease was associated with CAPA (p=0.02) but overall lung involvement over time did not differ between both groups (p=0.29). Regardless of time points, bronchial  wall thickening was associated with CAPA (p=0.03). Use of immunomodulating drugs during COVID-19 disease may be a risk factor for developing CAPA (p=0.04), due to low  numbers, a general conclusion has to be drawn with care. Please underline the clinical implication of the study.
  6. Conclusions L 380-386. In conclusion, patients with COVID-19 and pulmonary aspergillosis demonstrated a tendency to exhibit consolidation as main pattern in early COVID-19 as well as bronchial  wall thickening and may assist radiologists in raising suspicion of the disease. Chronolog-ical change of the abnormal extent on CT did not differ between both groups. Higher patient age and prior use of immunosuppressive drugs may be a risk factor. Previously reported findings for COVID-19 associated aspergillosis or invasive pulmonary aspergillosis (nodularity, halo sign, reverse halo sign or cavitating disease) were not observed. Please improve this paragraph and underline the novelty of this study.

Author Response

Dear editor, dear reviewer

We thank the reviewers for their comments and suggestions and found these to be helpful in clarifying our manuscript entitled “Clinical and imaging features of COVID-19 associated pulmonary aspergillosis” and conveying our results. We have edited our paper according to your suggestions and have listed below the changes made in response to each comment. The original comment is listed plain font and our responses are listed below each comment in a cursive typeface in in our point-to-point answer.

Reviewer: 1

  1. Introduction L37-42. The COVID-19 pandemic caused by SARS-CoV-2 is still challenging health-care systems due to new virus mutations with increased transmissibility and/or immune escape [1]. A large proportion of hospitalized patients require prolonged hospitalization and intensive care treatment. The case fatality ratio (defined as the proportion of individuals diagnosed with disease who die from that disease) is reported as 1-3% with higher risk among older patients with cardiovascular or metabolic diseases and men in general [2].  Could you please improve this paragraph and add this references:

a-Pelosi, P., Tonelli, R., Torregiani, C., Baratella, E., Confalonieri, M., Battaglini, D., Marchioni, A., Confalonieri, P., Clini, E., Salton, F., Ruaro, B. (2022). Different Methods to Improve the Monitoring of Noninvasive Respiratory Support of Patients with Severe Pneumonia/ARDS Due to COVID-19: An Update. Journal of clinical medicine11(6), 1704. https://doi.org/10.3390/jcm11061704

b-Asadi, F., Shahnazari, R., Bhalla, N., & Payam, A. F. (2021). Clinical evaluation of SARS-CoV-2 lung HRCT and RT-PCR Techniques: Towards risk factor based diagnosis of infectious diseases. Computational and structural biotechnology journal19, 2699–2707. https://doi.org/10.1016/j.csbj.2021.04.058

According to the reviewer’s suggestion, the mentioned paragraph was rewritten and improved, especially. The suggested references were added. The introduction of the manuscript now states the following: “The COVID-19 pandemic caused by SARS-CoV-2 has challenged health-care systems due to the introduction of a new virus without population immunity and subsequently new viral mutations with increased transmissibility and/or immune escape [1]. Already during the first wave, a large proportion of hospitalized patients required prolonged hospitalization and intensive care treatment. COVID-19 is a multisystem disorder with a complex pathophysiology [2]. In the lung SARS-CoV-2 causes diffuse alveolar damage. Particularly beyond the early viral replication phase, which is characterised by a viral syndrome, lung damage in severe COVID-19 is caused by immunothrombosis [2]. In acute respiratory failure, non-invasive support with close monitoring is now widely accepted as opposed to early intubation at the pandemic onset [3]. Increased risk for a fatal outcome includes older age, male gender, immunosuppression and cardiovascular or metabolic diseases [4]. Chest computed tomography has quickly become a standard complementary technique alongside rt-PCR for establishing a diagnosis and for risk assessment [5].”

  1. L48-49. Use of high-doses corticosteroids in patients hospitalized with COVID-19 has been shown to have a positive effect on patient survival but the immunosuppressant effect  leads to increasing superinfection by Aspergillus spp (COVID-19 associated aspergillosis, CAPA) [6,7].

Please improve this paragraph and add these references:

a-Chong WH, Neu KP. Incidence, diagnosis and outcomes of COVID-19-associated pulmonary aspergillosis (CAPA): a systematic review. J Hosp Infect. 2021 Jul;113:115-129. doi: 10.1016/j.jhin.2021.04.012. Epub 2021 Apr 21. PMID: 33891985; PMCID: PMC8057923.

b-Baratella E, Roman-Pognuz E, Zerbato V, Minelli P, Cavallaro MFM, Cova MA, Luzzati R, Lucangelo U, Sanson G, Friso F, Bussani R, Pinamonti M, Busetti M, Salton F, Confalonieri M, Ruaro B, Di Bella S. Potential links between COVID-19-associated pulmonary aspergillosis and bronchiectasis as detected by high resolution computed tomography. Front Biosci (Landmark Ed). 2021 Dec 30;26(12):1607-1612. doi: 10.52586/5053. PMID: 34994174.

Again, thank you for this valuable comment. The suggested references were added and the paragraph was rewritten. The introduction of the manuscript now states the following: “Use of high-doses corticosteroids in patients hospitalized with COVID-19 has been shown to have a positive effect on patient survival but the immunosuppressant effect leads to increasing risk of superinfection by Aspergillus spp (COVID-19 associated aspergillosis, CAPA) [9,10], which is reported with an overall incidence among COVID-19 patients of 13.5% (range 2.5 to 35.0%) [11]. Consensus criteria for case definition of CAPA have been proposed by the European Confederation for Medical Mycology (ECMM) and the International Society for Human and Animal Mycology (ISHAM). Since there is no established consensus for prophylaxis, early recognition of CAPA is essential for patient management and better radiographic diagnosis therefore desirable.

Three different grades were proposed: possible, probable and proven CAPA [10]. CAPA is reported in 1% to 28% of intensive care unit (ICU) patients [12–15] with mortality rates between 44 and 71% [15–17].

Imaging features have mostly been reported case-based without control groups. Consolidations and nodules were described as predominant CT findings, whereas cavities and halo signs are rare in contrast to patients with neutropenia-associated invasive pulmonary aspergillosis [10,18–23]. A recent study found a potential association between CAPA and bronchiectasis [24].

Imaging features have mostly been reported case-based without control groups. Consolidations and nodules were described as predominant CT findings, whereas cavities and halo signs are rare in contrast to patients with neutropenia-associated invasive pulmonary aspergillosis [9,17–22]. A recent study found a potential association between CAPA and bronchiectasis [23].”

  1. L 59-61. Therefore, the aim of this study was to compare clinical and imaging features between CAPA patients and a well characterized group of critically ill COVID-19 patients without evidence of pulmonary aspergillosis. Please improve the description of study aim and underline the novelty of the study.

The aim of the study was rewritten as suggested by reviewer 1. The novelty was underlined. The aim of the study is now described as follows: “This is the first study, that aims to establish typical CAPA imaging features by comparing CT scans of CAPA patients to scans of COVID-19 patients without evidence of pulmonary aspergillosis.”

  1. 1. Imaging. Please summarise all the information regarding the imaging in the same paragraph.

As suggested, all information regarding imaging and image evaluation was summarized in one paragraph. The subheading 2.1. was renamed to “imaging and image evaluation”, the subheadings 2.2 to 2.5 were omitted.

  1. Table 1. Key demographic and clinical characteristics. Please add the most important clinical findings of patients.

Thank you for addressing this issue. The revised manuscript demonstrates the clinical data (e.g., preexisting conditions, prior medication, treatment during COVID-19 disease) in table 3, moreover table 1 was renamed exclusively to “Key demographics”

  1. Figure 3. L259-261. Imaging example of bronchial wall thickening in a patient with probable COVID-19 associated pulmonary aspergillosis (CAPA). CT was performed 10 days after symptom onset at the day of the CAPA diagnosis. T/D ratio (wall thickness (T) divided by the total diameter of bronchus (D) was 0.32 in this case. Please improve the quality of figure 3.

Thank you for addressing this issue, figure 3 was improved; resolution is 300dpi. The bronchus with the thickened wall is now more centered in the figure and the arrow was moved and enlarged to better depict the pathology.

  1. Discussion. L 297- 302. This case-control study compared clinical and imaging features between COVID-19 patients and patients with COVID-19 associated pulmonary aspergillosis (CAPA). CAPA  patients were older compared to COVID-19 patients (p=0.01). Consolidation in early  COVID-19 disease was associated with CAPA (p=0.02) but overall lung involvement over time did not differ between both groups (p=0.29). Regardless of time points, bronchial wall thickening was associated with CAPA (p=0.03). Use of immunomodulating drugs during COVID-19 disease may be a risk factor for developing CAPA (p=0.04), due to low  numbers, a general conclusion has to be drawn with care. Please underline the clinical implication of the study.

Thank you for this valuable comment. The clinical implications were underlined as suggested in the mentioned paragraph. The following sentence about clinical impact was added: “The imaging features described in this study support physicians in diagnosing CAPA and may lead to improved patient management.”

  1. Conclusions L 380-386. In conclusion, patients with COVID-19 and pulmonary aspergillosis demonstrated a tendency to exhibit consolidation as main pattern in early COVID-19 as well as bronchial  wall thickening and may assist radiologists in raising suspicion of the disease. Chronolog-ical change of the abnormal extent on CT did not differ between both groups. Higher patient age and prior use of immunosuppressive drugs may be a risk factor. Previously reported findings for COVID-19 associated aspergillosis or invasive pulmonary aspergillosis (nodularity, halo sign, reverse halo sign or cavitating disease) were not observed. Please improve this paragraph and underline the novelty of this study.

Thank you for pointing this out. According to your suggestion, we summarized the findings of this study with more precision. Moreover, the novelty of this study was emphasized. This conclusion now states the following: “In conclusion, patients with COVID-19 associated pulmonary aspergillosis showed a tendency for consolidation in early COVID-19 disease. Bronchial wall thickening and higher patient age were associated with CAPA. Prior use of immunosuppressive drugs may be a risk factor. Previously reported findings for COVID-19 associated aspergillosis or invasive pulmonary aspergillosis (nodularity, halo sign, reverse halo sign or cavitating disease) were not observed. This is the first study of its kind and findings may assist physicians in raising suspicion of the disease.”

Reviewer 2 Report

Dear the authors.

I am really grateful to be given this opportunity to review the article, entitled "Clinical and imaging features of COVID-19 associated pulmonary aspergillosis".

As one of the physicians to see patients with COVID-19, I've read the manuscript with great interest.

The study has been well planned and organized, resulting in the conclusion that higher patient age and prior use of immunosuppressive medications may be a risk factor.

I would like to place two comment for the authors.

First comment;

Are there any prophylactic tactics against CAPA or any precaution to detect CAPA earlier for treatment.

Second comment;

The authors describe that three different grades of CAPA (possible, probable, and proven).  In the present study, CAPA diagnosis was one case of proven, ten cases of probable, and two cases of possible.

I know it is difficult to examine the statistical difference between these subgroup, however, is there any difference regarding the clinical course of these three subtypes  (possible, probable, and proven)?

I hope my comments would be helpful for the authors.

Thank you.

Author Response

Dear editor, dear reviewer

We thank the reviewers for their comments and suggestions and found these to be helpful in clarifying our manuscript entitled “Clinical and imaging features of COVID-19 associated pulmonary aspergillosis” and conveying our results. We have edited our paper according to your suggestions and have listed below the changes made in response to each comment. The original comment is listed plain font and our responses are listed below each comment in a cursive typeface in in our point-to-point answer.

Reviewer: 2

Dear the authors.

I am really grateful to be given this opportunity to review the article, entitled "Clinical and imaging features of COVID-19 associated pulmonary aspergillosis".

As one of the physicians to see patients with COVID-19, I've read the manuscript with great interest.

The study has been well planned and organized, resulting in the conclusion that higher patient age and prior use of immunosuppressive medications may be a risk factor.

Thank you, we appreciate the positive assessment of our study.

I would like to place two comment for the authors.

First comment;

Are there any prophylactic tactics against CAPA or any precaution to detect CAPA earlier for treatment.

Thank you for pointing this out. This is still an area of uncertainty as there is no consensus for prophylaxis. Therefore, early detection of CAPA is essential for management. We have added the following statement: “Since there is no established consensus for prophylaxis, early recognition of CAPA is essential for patient management and better radiographic diagnosis therefore desirable.”

Second comment;

The authors describe that three different grades of CAPA (possible, probable, and proven).  In the present study, CAPA diagnosis was one case of proven, ten cases of probable, and two cases of possible.

I know it is difficult to examine the statistical difference between these subgroup, however, is there any difference regarding the clinical course of these three subtypes (possible, probable, and proven)?

I hope my comments would be helpful for the authors.

Thank you.

Thank you for mentioning this issue. We did a subgroup analysis and compared the different CAPA groups (possible, probable and proven) with regard to the following variables: outcome (survived / died), time from symptom onset to proof of COVID-19 (days) and time from symptom onset to proof of CAPA (days) with the following results:

Possible CAPA (n=2)

Probable CAPA (n=10)

Proven CAPA (n=1)

Outcome

1 (50%)

6 (60%)

1 (100%)

Symptom onset to COVID-19 (median [IQR])

6.50 [4.75, 8.25]

6.50 [1.50, 9.75]

0.00 [0.00, 0.00]

Symptom onset to CAPA (median [IQR])

16.00 [14.00, 18.00]

16.00 [13.50, 30.75]

0.00 [0.00, 0.00]

We consulted our statistical expert, who is a co-author of this study, regarding statistical tests for this comparison. Her answer was, that due to the low number especially in the “proven” group (n=1), test for statistically significant differences would not be contributing. By comparing the groups (especially the “possible” and the “probable” group) with “the naked eye” we did not see apparent differences. We therefore chose not to include this table into the manuscript at this time. Upon editorial request we are of course prepared to add this table to the final article.

Round 2

Reviewer 1 Report

All comments have been solved by authors, and the manuscript has been substantially improved.
No further comments